# No effect of monetary reward in a visual working memory task

**Ronald van den Berg** [1]*, **Qijia Zou**[2], **Yuhang Li**[3], **Wei Ji Ma**[4]

**1** Department of Psychology, Stockholm University, Stockholm, Sweden, **2** Department of Psychology, New York University, New York, New York, United States of America, **3** Centre for Cognitive and Brain Sciences, University of Macau, Taipa, Macao, **4** Department of Psychology, Center for Neural Science, New York University, New York, New York, United States of America

* ronald.van-den-berg@psychology.su.se, ronald.vandenberg@psyk.uu.se

## Abstract

Previous work has shown that humans distribute their visual working memory (VWM) resources flexibly across items: the higher the importance of an item, the better it is remembered. A related, but much less studied question is whether people also have control over the *total* amount of VWM resource allocated to a task. Here, we approach this question by testing whether increasing monetary incentives results in better overall VWM performance. In three experiments, subjects performed a delayed-estimation task on the Amazon Turk platform. In the first two experiments, four groups of subjects received a bonus payment based on their performance, with the maximum bonus ranging from $0 to $10 between groups. We found no effect of the amount of bonus on intrinsic motivation or on VWM performance in either experiment. In the third experiment, reward was manipulated on a trial-by-trial basis using a within-subjects design. Again, no evidence was found that VWM performance depended on the magnitude of potential reward. These results suggest that encoding quality in visual working memory is insensitive to monetary reward, which has implications for resource-rational theories of VWM.

## Introduction

A central question in research on human visual working memory (VWM) is how much flexibility exists in how the system distributes its resource across encoded items [1, 2]. The answer to this question partly depends on how one conceptualizes the nature of VWM resource. One class of models postulates that VWM consists of a small number of "slots" that each provide an indivisible amount of encoding resource (e.g., [3–7]). Since the number of slots is typically assumed to be very small (3 or 4), these models allow for virtually no flexibility in resource allocation. A competing class of models conceptualizes VWM as a continuous resource (e.g., [8–13]), sometimes in combination with a limit on the number of encoded items [14, 15]. Since a continuous resource can be divided into arbitrarily small packages, these models allow for a high degree of flexibility in resource allocation.

Several recent studies have found evidence for flexibility in VWM resource allocation. First, it has been found in multiple experiments that when one item in a stimulus array is more likely

**Data Availability Statement:** All data are available at https://osf.io/mwz27/.

**Funding:** This research was supported by grant 2018-01947 from the Swedish Research Council to R.v.d.B, training grant R90DA043849-03 from

National Institutes of Health to Q.Z., and grant R01EY020958-09 from National Institutes of Health to W.J.M. The funders had no role in study design, data collection and analysis, decision to publish, or preparation of the manuscript. There was no additional external funding received for this study."

**Competing interests:** The authors have declared that no competing interests exist.

to be selected for test than other items (i.e., have a higher "probing probability"), subjects remember this item with better precision [16–21]. Similarly, people remember items associated with a higher reward better than items associated with a lower reward [22]. In addition, it has been reported that subjects can make a tradeoff between the number of items in VWM and the quality with which they are encoded [23] (however, see [24]). By allocating more resources to the more important items within a display, subjects in these studies increased their performance compared to what it would have been if they had encoded all items within each display with the same precision. This suggests that VWM resource allocation may be driven by a rational policy.

We recently formalized this suggestion by modeling VWM as a rational system that balances the amount of invested resource against expected task performance: the more there is at stake, the more resource is allocated for encoding [25]. This "resource-rational" interpretation of VWM predicts two kinds of flexibility in the allocation of VWM resource. First, items of unequal importance are assigned unequal amounts of encoding resource. For example, when one item in a memory array is more likely to be probed, the resource-rational strategy would be to encode it with higher precision than the other items. In support of this prediction, we found that the model provided excellent quantitative fits to data from previous experiments that varied probing probabilities [19, 20]. A second prediction made by the model is that *tasks* of unequal importance are assigned unequal amounts of *total* resource: the higher the incentive to perform well on a task, the more VWM resource a subject should be willing to invest. In support of the second kind of flexibility, it has been found that subjects who are encouraged to "try to remember all items" in a change detection task have higher estimated numbers of slots than subjects who are told to "just do your best" or to "focus on a subset" [26]. Moreover, there is evidence that people can flexibly trade off resources between auditory and visual working memory based on the amount of reward associated with each task [27–29]. Finally, there are indications that retro-cueing can increase net VWM capacity [30].

In the present study, we examine whether the precision with which people encode a set of stimuli in VWM depends on the amount of monetary reward they get for good performance. We performed three experiments in which subjects earned a performance-contingent monetary bonus on top of a base payment. When encoding is costly, a rational observer should adjust its total amount of invested VWM resource to the amount of performance-contingent bonus: the higher the potential bonus, the more effort should be put into the task. We did not find evidence for such an effect in any of the experiments. In opposition to the prediction following from a resource-rational theory of VWM [25], the present results suggests that encoding precision in VWM is insensitive to monetary reward.

## Experiment 1

### Recruitment

Subjects were recruited on the Amazon Mechanical Turk platform, where the experiment was posted as a "Human Intelligence Task". The experiment was visible only to subjects who were located in the USA, had not participated in the experiment before, and had an approval rate of 95% or higher. A total of 355 subjects signed up, of which 156 were disqualified due to failing the post-instruction quiz (see below). The remaining 199 subjects were randomly assigned to four groups ($n$ = 49, 47, 47, 46) that differed in the total amount of bonus they could earn by performing well (no bonus, $2, $6, $10). Besides the bonus, subjects received a $1 base payment. Hence, subjects in the groups with a bonus could increase their total earnings by a factor of 3, 7, or 11. The experiment was approved by the Institutional Review Board of New York University.

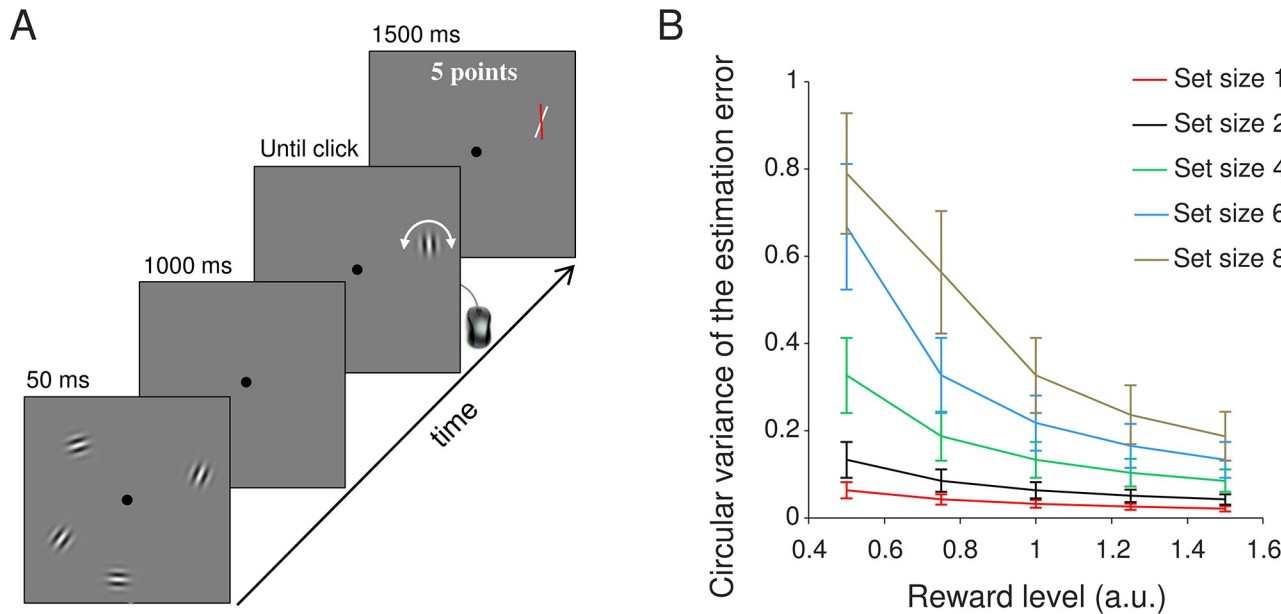

**Fig 1. Experimental procedure.** (A) Illustration of a single trial in Experiment 1 (not to scale). Subjects were briefly presented with 1, 2, 4, 6, or 8 Gabor patches, which they had to keep in memory during the delay period. Thereafter, a randomly oriented Gabor patch would appear at one of the previous stimulus locations. The task was to match the orientation of this stimulus with the remembered orientation of the stimulus that had appeared earlier at this location. After submitting their response, the subject was presented with feedback about the size of their response error. Subjects received points on each trial (larger error = fewer points), which after the experiment were converted into monetary reward. Different groups of subjects received different levels of reward. (B) Predictions of the resource-rational model about the effect of reward level (*x* axis) on VWM error *(y* axis). The predictions were obtained by simulating responses of the model presented in Van den Berg & Ma (2018). Simulations were performed at five set sizes (separate lines) and 5 reward levels. Each of the simulations was performed six times, with each run using the maximum-likelihood parameters of one of the six subjects in experiment E4 Van den Berg & Ma (2018). Error bars represent ±1 SEM across the six runs. A two-way Bayesian ANOVA on the simulation data show strong evidence for an effect of both set size ($BF_{incl} > 10^{13}$) and reward level ($BF_{incl} > 10^8$), as well as for an interaction effect ($BF_{incl} = 98$).

## Stimuli and task

On each trial, the subject was presented with 1, 2, 4, 6, or 8 Gabor patches, which were placed along an invisible circle around a central fixation point (Fig 1A). We refer to the number of presented items as the set size, which varied from trial to trial in a pseudo-random manner. The orientation of each patch was drawn independently from a uniform distribution over all possible orientations. The stimulus appeared for 50 milliseconds and was followed by an empty screen with a duration of 1 second (memory period). Thereafter, a randomly oriented Gabor patch appeared at one of the previous stimulus locations, whose initial orientation was randomly drawn and could be adjusted through mouse movement. The task was to match the orientation of this probe stimulus with the remembered orientation at that location. Only one item was probed on each trial. After submitting the response, the error between the correct orientation and the reported orientation, $\varepsilon$, was converted into an integer score between 0 and 10, with more points assigned for smaller errors (see S1 Appendix for a visualization of the scoring function). Feedback was provided after each trial by showing the obtained score and two lines that corresponded with the correct and responded orientations.

## Procedure

At the start of the experiment, subjects received written instructions about the task and about how their performance would be scored. Next, they were informed about the bonus payment. For a subject in the condition with a maximum bonus of $10, the text in this screen would

read "*After the experiment, we will randomly select THREE trials and use the HIGHEST score among the three to decide how much bonus you will receive [. . .] You will receive $1 for each point you get on a trial. For example, if your highest score among the three trials is 7 points, then your actual bonus is $7. You will not receive any bonus if you get 0 points*!". Thereafter, they performed 15 practice trials that were identical to trials in the actual experiment. After finishing these trials, a multiple-choice quiz was presented with three questions to test the subject's understanding of the task and the potential bonus payment. Subjects who failed on at least one of these questions were disqualified from the experiment. The remaining subjects performed 250 trials of the delayed-estimated task with the five set sizes pseudo-randomly intermixed. To check if subjects were paying attention, we asked them at three points in the experiment to press the space bar within 4 seconds (catch trials). Subjects who at least once failed to do this were presumably not paying attention and were therefore excluded from the analyses.

## Model predictions

To get an idea of the kind of data we may expect if the predictions of the resource-rational model are correct, we performed a simulation analysis. The results (Fig 1B) show that the estimation error (quantified as the circular variance of the estimation distribution) is predicted to decrease as the amount of potential reward increases. Moreover, this effect is predicted to be most pronounced at larger set sizes.

## Results

Data from 10 subjects were excluded from the analyses because they failed to respond to at least one of the three catch trials. Of the remaining 189 subjects, another 35 were excluded because they had response error distributions that did not significantly differ from a uniform distribution, as assessed by a Kolmogorov-Smirnov test with a significance level of 0.05. The final group on which we performed the analysis thus consisted of 154 subjects, with 37, 41, 38, and 38 in the $0, $2, $6, and $10 conditions, respectively. To test for effects of reward level on VWM performance, we computed for each subject the circular variance of the response error distribution at each set size (Fig 2A, left). We performed a Bayesian repeated-measures ANOVA [31, 32] on these measures, with set size as a within-subjects factor and bonus level as a between-subjects factor. We express the evidence in favor of an effect as the 'inclusion Bayes Factor' ($BF_{incl}$), which measures how likely the data are in models that include the factor compared to how likely they are in models that exclude the factor. Likewise, evidence *against* an effect (i.e., in favor of the null hypothesis) will be expressed as the 'exclusion Bayes Factor' ($BF_{ecxl}$), which measures how likely the data are in models that exclude the factor compared to how likely they are in models including the factor. The results of the ANOVA indicated extremely strong evidence for a main effect of set size ($BF_{incl} = \infty$), but evidence *against* a main effect of bonus level ($BF_{excl} = 21$).

## Discussion

The results of Experiment 1 showed no evidence of an effect of performance-contingent reward on VWM performance. One possible explanation of this null result is that resource allocation in VWM is insensitive to monetary reward. However, there are at least two factors in the experimental design that may have interfered with the reward manipulation. First, subjects received trial-to-trial feedback. Being constantly confronted with their own performance may have motivated them to perform as well as possible regardless of the amount of bonus they could earn. Second, since the bonus was mentioned only at the beginning of the experiment, subjects may have performed the task without having the bonus strongly on their

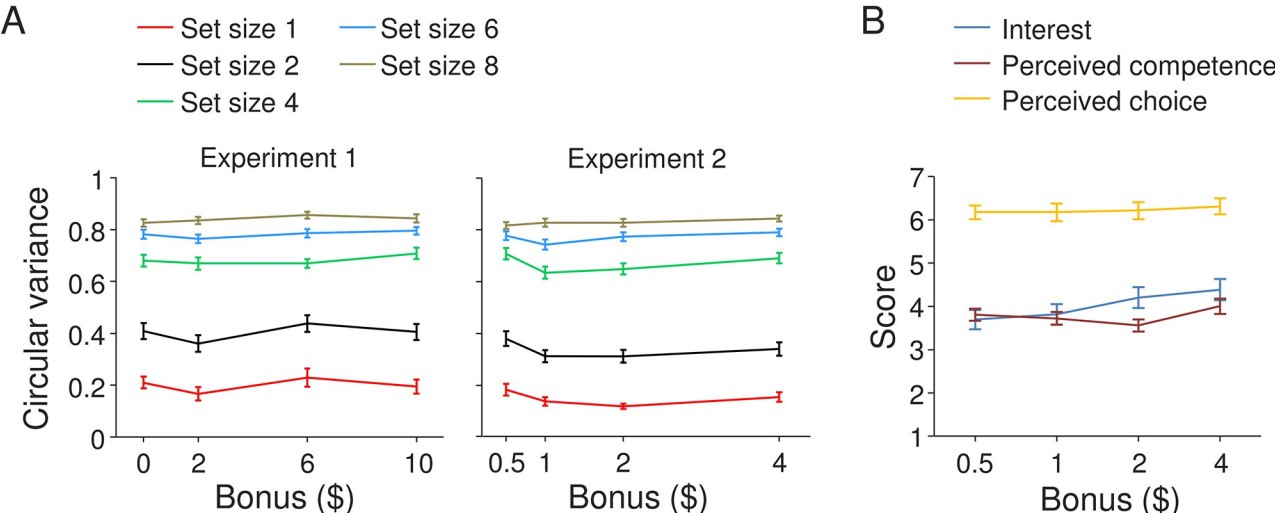

**Fig 2. Effect of monetary reward level on memory precision and self-reported motivation scores in Experiments 1 and 2.** (A) Subject-averaged circular variance of the estimation error distribution as a function of the amount of potential bonus in Experiments 1 (left) and 2 (right). (B) Intrinsic Motivation Inventory scores as a function of the amount of potential bonus, split by item category. Error bars indicate 1 s.e.m.

minds. To address these potential confounds, we ran a second experiment in which subjects did not receive trial-to-trial feedback and were reminded regularly of the bonus.

## Experiment 2

### Recruitment

A new cohort of subjects was recruited on the Amazon Mechanical Turk platform. The experiment was visible only to subjects who were located in the USA, had not participated in the experiment before, and had an approval rate of 95% or higher. A total of 241 subjects signed up, of whom 41 were disqualified due to failing the post-instruction quiz. The remaining 200 subjects were randomly assigned to four groups ($n$ = 52, 48, 50, 50) that again differed in the amount of potential bonus payment. The base payment was $5 and the potential bonus amounts were $0.50, $1, $2, and $4. Hence, subjects in the groups with a bonus could increase their total earnings by 10%, 20%, 40% or 80% if they performed well. The experiment was approved by the Institutional Review Board of New York University.

### Stimuli and procedure

The stimuli and procedure for Experiment 2 were identical to Experiment 1, except for the following differences. First, subjects were reminded of the bonus four times in the instruction screen (compared to only once in Experiment 1) and during the task itself the following message appeared after every 50 trials: "*You have completed X% of the Experiment. Remember that you have the chance to earn a $Y bonus!*", where $X$ and $Y$ were determined by the number of completed trials and the amount of bonus, respectively. Second, no performance feedback was given, neither during practice nor during the actual experiment. Third, the length of the practice phase was reduced to 10 trials, but three "walk-through trials" were added at the start in which subjects were fully guided with additional written instructions. Lastly, after the experiment, subjects rated 20 items that we had selected from the Intrinsic Motivation Inventory [33, 34]. We only included the subscales "Interest/Enjoyment" (7 items, such as "This activity

was fun to do"), "Perceived competence" (6 items, such as "I was pretty skilled at this activity"), and "Perceived Choice" (7 items, such as "I did this activity because I wanted to") in our questionnaire. Subjects rated these items on a Likert scale from 1 ("not at all true") to 7 ("very true"). An overview of all items is found in the S1 Appendix.

## Results

Data from 27 subjects were excluded because they failed to respond to one of the catch trials (9 subjects) or had a response error distribution that did not significantly differ from a uniform distribution according to a Kolmogorov-Smirnov test (18 subjects). After these exclusions, we had 47, 42, 44, and 40 subjects in \$0.50, \$1, \$2, and \$4 conditions, respectively. We performed the same statistical analyses as in Experiment 1 on the data from the remaining 173 subjects (Fig 2A, right). Again, we found extremely strong evidence for a main effect of set size ($BF_{incl}$ = $\infty$) and evidence *against* a main effect of bonus level ($BF_{excl}$ = 2.9). Hence, it seems unlikely that the absence of an effect in Experiment 1 was due to subjects being unaware of the potential bonus payment or due to presence of trial-to-trial feedback.

Next, we assessed whether bonus size affected the subjects' scores on the intrinsic motivation inventory questions (Fig 2B). Using Bayesian one-way ANOVAs, we found that there was no effect in any of the three categories: $BF_{10}$ = 0.275 for mean "interest" scores, $BF_{10}$ = 0.174 for mean "perceived competence" scores, and $BF_{10}$ = 0.034 for mean "perceived choice" scores. Nevertheless, we noticed that there was considerable variation in the intrinsic motivation scores across subjects, especially in the "Interest" and "Perceived competence" categories (Fig 3A). Therefore, we next tested if there was an effect of motivation scores on VWM performance. To this end, we grouped subjects from Experiment 2 into "low motivation" and "high motivation" subgroups by using a median split on each of the three categories of the Intrinsic Motivation Inventory (Fig 3B). To examine whether scores in any of the three categories is predictive of VWM performance, we performed a repeated-measures Bayesian ANOVA with set size as within-subjects factor and motivation score ("low" and "high") as a between-subjects factor. All three tests provided evidence for the null hypothesis that there was no performance difference between subjects in the low and high motivation subgroups (Interest: $BF_{excl}$ = 8.3; Perceived competence: $BF_{excl}$ = 4.8; Perceived choice: $BF_{excl}$ = 4.8).

## Discussion

The aim of Experiment 2 was to test whether the null effect from Experiment 1 persists if we remove trial-by-trial feedback and remind subjects more often of the potential bonus. We found that this was the case: again, there was no effect of monetary reward on VWM performance. This further strengthens the hypotheses that VWM resource allocation is independent of monetary reward. We also found that intrinsic motivation did not depend on the amount of monetary reward. Therefore, our present results are limited to the realm of external motivation manipulations. It would be useful to also test for effects in a design where intrinsic motivation is experimentally manipulated.

In both Experiment 1 and Experiment 2, we manipulated reward using a between-subjects design. Hence, for each subject the expected reward was the same for each item on each trial. However, there are previous indications that effects of monetary reward on cognitive performance are *relative* rather than absolute, which can be detected using within-subject designs in which reward is manipulated trial-by-trial (e.g., [35–42]). To test whether VWM encoding quality is affected by relative reward levels, we performed a third experiment in which reward was varied on a trial-by-trial level.

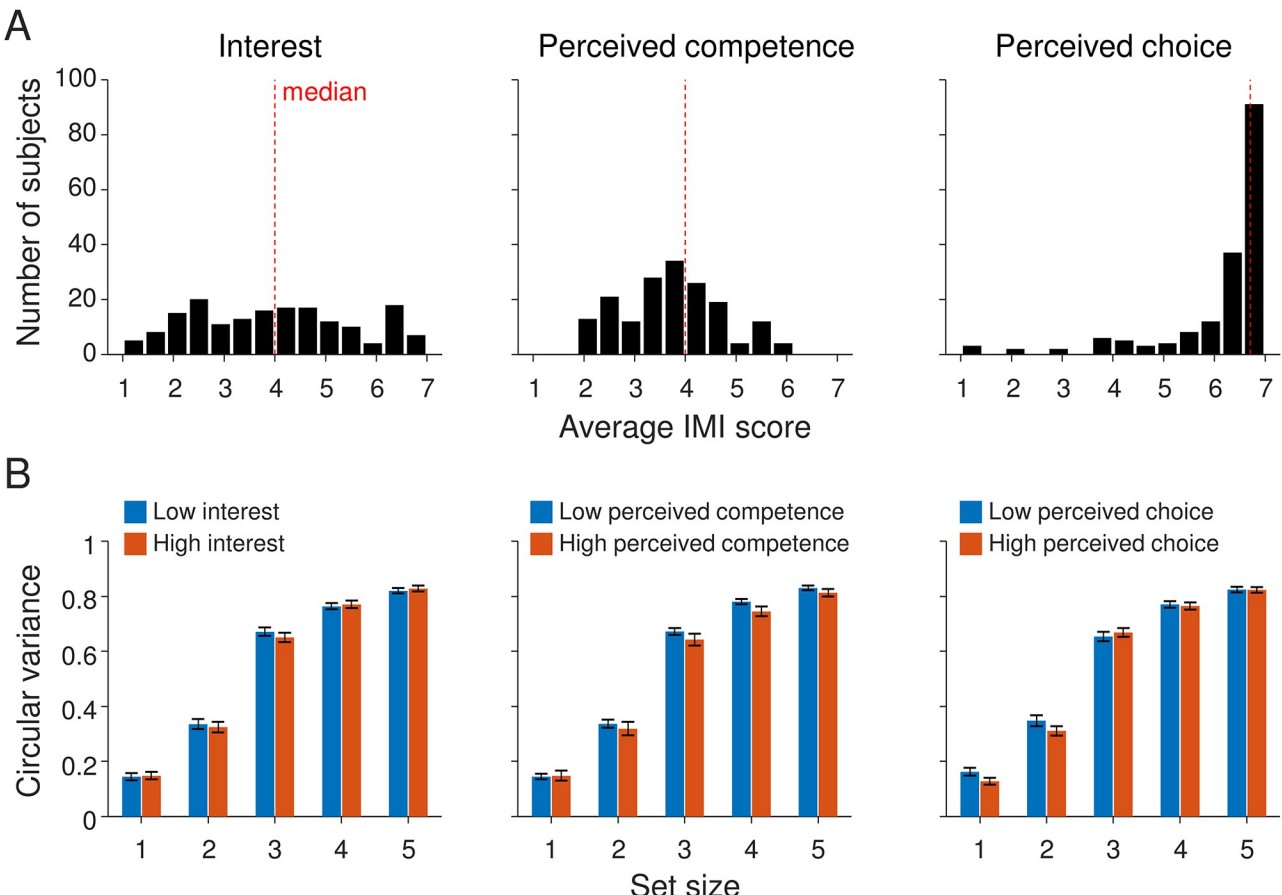

**Fig 3. Comparison of VWM performance between subjects with low and high scores on the Intrinsic Motivation Inventory (IMI).** (A) Distribution of average IMI scores, split by question category. (B) Circular variance of the response error plotted separately for subjects with below-median and above-median scores on the IMI questionnaire.

## Experiment 3

### Preregistration

We preregistered the methods of Experiment 3 at the Open Science Foundation platform (URL: https://osf.io/dkzax). The report below is in accordance with the preregistration, except when explicitly indicated otherwise.

### Recruitment

A third cohort of subjects was recruited on the Amazon Mechanical Turk platform. The experiment was visible only to subjects who were located in the USA, had not participated in the experiment before, and had an approval rate of 95% or higher. Data was collected from a total of 201 subjects, which were randomly assigned to two groups that differed in how the final payment was calculated (see below for details). The base payment was $8 and subjects could earn a bonus up to $10.50. Hence, subjects could increase their total earning by approximately a factor 2.3 if they performed well.

## Stimuli and procedure

Experiment 3 was similar to Experiments 1 and 2, but with a few important differences. Just as in the first two experiments, the instructions at the start of the experiment informed subjects that they would score points on each trial and that their score would be converted into a monetary bonus. Subjects received a score between 0 and 10 points on each trial, depending on how close their response was to the correct response. The mapping between the absolute error, $\varepsilon$ (in degrees), and the score, $s$, was similar as in Experiments 1 and 2 (see S1 Appendix). In the instructions, this mapping was visualized to the subject by showing an example error corresponding to each of the 11 possible scores, similar to the way feedback was provided in Experiment 1 (Fig 1A). Subjects were also informed that each trial had a *score multiplier* that could take values 1, 2, and 4. The multiplier was visualized as a text "1x", "2x", or "4x" at the center of the screen. To ensure that subjects would not forget about the multiplier, the text stayed on the screen throughout the entire trial (it also served as a fixation marker). The subject's response score was on each trial multiplied by the trial's score multiplier. At the end of the trial, the error was visualized using two lines (Fig 1A) and feedback about the gained score was provided by showing "[multiplier] x [response score] = [multiplied score]" at the fixation location (e.g., "2 x 6 = 12").

After reading the instructions, subjects were presented with three walk-through trials (one for each multiplier). These self-paced trials explained once again the relation between the response error and the score as well as the role of the score multiplier. Next, they performed 18 practice trials that were identical to the experimental trials, but which were not included in the analyses. Thereafter, they were presented with three multiple-choice questions to check if they had understood the task and the way that their payment was calculated. Subjects who failed to answer all three questions correctly, were disqualified from performing the experiment. Subjects who passed the quiz would next start the main experiment, which consisted of 40 trials for each combination of set size (1, 3, or 6) and multiplier (1x, 2x, 4x), presented in random order (360 trials in total). Note that here we reduced the number of trials from 50 (in the preregistration) to 40, which was done to keep the experiment duration to approximately an hour. After performing the experiment, subjects were asked to indicate on a scale from 1 to 5 how much effort they you put in trials with a 1x, 2x, and 4x score multiplier (they provided three ratings, one for each multiplier).

## Two payment conditions (between-subject factor)

Since we measure hundreds of trials per subject, the expected reward per trial is quite low. A risk of this is that at the level of a single trial, subjects may perceive the expected bonus as so low that it is not really different from getting no bonus at all. Therefore, we divided subjects into two groups which differed in how their monetary bonus was calculated. In the first group, the bonus was calculated based on the summed score across *all* trials, with every 8 points being worth 1 cent. In the second group, the bonus was based on the total score of 9 trials (3 for each multiplier) that would randomly be selected after the experiment, with each 20 points being worth $1. These numbers were chosen such that the maximum total reward as well as the expected reward under guessing were the same in both groups. However, subjects in the second group may have felt that there was more at stake on any given trial than subjects in the first group ("what if this is one of the bonus trials?"). Note that the bonus calculations differed slightly from how we had formulated them in the preregistration ("Every 15 points in your total multiplied scores is worth 2 cents" for the all-trials group and "Every 15 points in your total multiplied score is worth $1" for the nine-random-trials group), but the underlying idea was still the same.

## Results

A total of 52 subjects were excluded because they failed to respond to one or more of the catch trials. Another 49 subjects were excluded because a Kolmogorov-Smirnov test failed to reject the hypothesis (at a $p < 0.05$ level) that their response error distribution was uniform, suggesting that they submitted random responses. The analyses reported below were performed on the 100 remaining subjects (with 52 and 48 in the first and second payment group, respectively). Unfortunately, due to a technical error, the post-experiment questionnaire data were collected for only a few of the subjects; we did not use these data in any of our analyses.

We performed a Bayesian ANOVA with performance (measured as the circular variance of the response error) as the dependent variable, multiplier and set size as within-subject factors, and payment condition as a between-subject factor. The results revealed extremely strong evidence for a set size effect ($BF_{incl} > 10^{13}$), but no evidence for an effect of payment condition ($BF_{excl} = 9.0$) or multiplier ($BF_{excl} = 38$) (Fig 4).

While not specified in the preregistration, we also performed a linear mixed modelling analysis using the *lme4* package for R [43]. This analysis is possibly more powerful in detecting effects, because it can deal with individual differences in baseline performance. Set size, multiplier, and payment condition were specified as fixed effects and the intercept for subjects as a random effect. To estimate the evidence for an effect, we performed a Chi-square test between the full model and the model with the effect removed. Consistent with the ANOVA, this provided strong evidence for an effect of set size ($p < 0.0001$), but no evidence for an effect of multiplier ($p = 0.14$) or payment condition ($p = 0.56$).

Next, we analyzed the data at the level of individuals by performing the model-based analysis as planned in the preregistration. For each subject, we first fitted a variable-precision model [10, 14] in which mean encoding precision, $\bar{J}$, depended only on set size, $N$, through the power-law relation $\bar{J}(N) = \bar{J}_1 N^\alpha$, where parameter $\bar{J}_1$ determined the precision at set size 1 and parameter $\alpha$ how fast precision decreases with set size, respectively. The third parameter of this model, $\tau$, determines the amount of variability in encoding precision. We compared the goodness of fit of this model to a variant in which encoding precision also depended on the

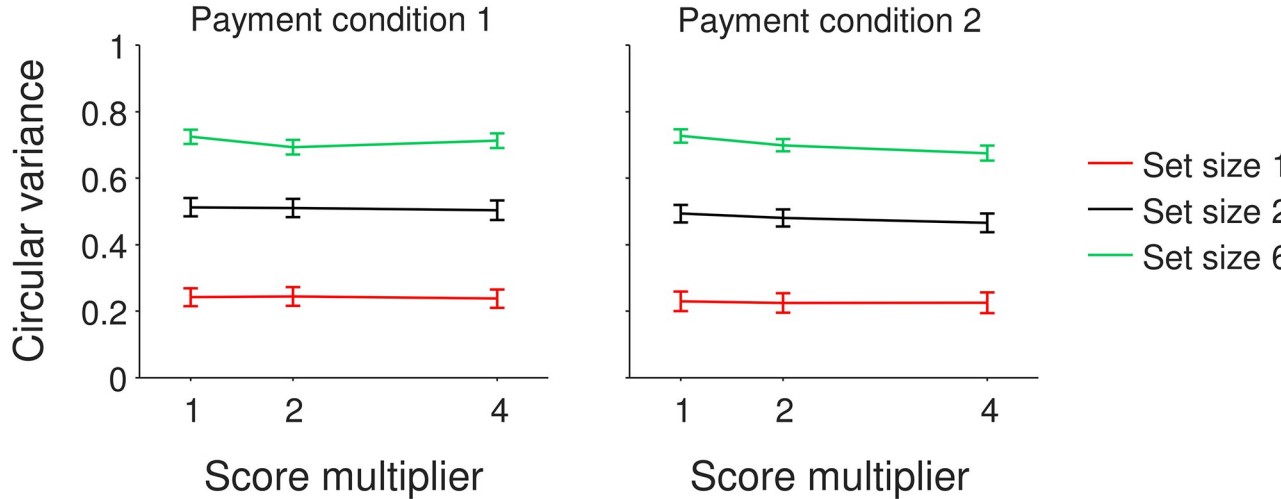

**Fig 4. Effect of monetary reward level on memory precision in Experiment 3.** (A) Subject-averaged circular variance of the estimation error distribution as a function of the trial-by-trial score multiplier. Payment Condition 1 refers to the group of subjects whose bonus was calculated on the score summed across all trials and Payment Condition 2 to the group whose bonus was calculated on 9 randomly selected trials. Error bars indicate 1 s. e.m.

**Table 1. Maximum-likelihood parameters of the two models fitted to the data of Experiment 3.**

| Model | Standard VP | | | Flexible VP | | | | | | |
|---|---|---|---|---|---|---|---|---|---|---|
| Parameter | $\bar{J}_1$ | $\alpha$ | $\tau$ | $\bar{J}_{1,1}$ | $\bar{J}_{1,2}$ | $\bar{J}_{1,4}$ | $\alpha_1$ | $\alpha_2$ | $\alpha_4$ | $\tau$ |
| Mean | 6.75 | -1.21 | 9.5 | 6.79 | 7.02 | 7.19 | -1.54 | -1.41 | -1.34 | 13.6 |
| Standard error | 0.48 | 0.05 | 3.4 | 0.51 | 0.65 | 0.60 | 0.12 | 0.11 | 0.09 | 7.7 |

VP stands for 'Variable Precision'.

reward level, by fitting $\bar{J}_1$ and $\alpha$ separately for trials with multipliers 1, 2, and 4 (thus introducing four additional parameters). The second model has an AIC value that is on average only 3.11 ± 0.21 points lower than that of the first model, which indicates that the models describe the data approximately equally well. This result is corroborated by the parameter estimates (Table 1), which in the 7-parameter model are nearly identical across the three multipliers. Indeed, Bayesian repeated-measures ANOVA tests provide strong evidence in favor of the hypothesis that the parameter estimates are independent of multiplier ($BF_{excl}$ = 18.2 for $\bar{J}_1$; $BF_{excl}$ = 10.1 for $\alpha$). In conclusion, the model comparison result is consistent with the above statistical analyses and indicates that there is no evidence for an effect of reward level on encoding precision.

## Discussion

As in the first two experiments, we did not find any indications for an effect of monetary reward level on VWM performance. The main difference with the first two experiments is that Experiment 3 manipulated relative rather than absolute reward, by varying the potential monetary bonus from trial to trial. Another new manipulation in this experiment was the way in which bonus was calculated: for half the subjects the bonus was based on *all* trials while for the other half it was based on 9 randomly selected trials. We found no effect of this manipulation either on VWM precision, which makes it less likely that the null effects are due to subjects perceiving the per-trial bonus as too low to have any effect on their motivation.

## General dicsussion

Previous studies have found that people have some degree of control over how they allocate VWM resource across items: when experimental subjects are told that particular items are more important or valuable to remember than others, they encode these with higher quality than those that are believed to be less important or valuable [16–22]. Here we asked a related but less studied issue: is the *total amount* of VWM resource utilized to a task also under control of the participant? Assuming that the use of cognitive resources is costly, this kind of flexibility is predicted if VWM is a 'resource rational' system [25, 44]: the more there is at stake, the more resources the brain should put into performing well. We investigated this in three experiments in which we varied the values of task items through manipulations of monetary reward. In all three of them, we found no evidence that VWM capacity was sensitive to the experimental manipulations. In fact, the evidence consistently was in favour of the null hypothesis of there being *no* effect of monetary stake on VWM capacity.

Our results limit the generality of the resource-rational theory of VWM, but they do not imply a full refutation. As we discussed in our original publication of that theory [25], resource-rational optimization can take place at multiple timescales. Therefore, it could be that VWM allocation is rational only at timescales different from the ones tested in the experiments presented here. Moreover, while it is commonly assumed that monetary reward increases

motivation, there are also studies suggesting that reward may have very little [45] or even detrimental effects on motivation [46] and cognitive control mechanisms [47]. Therefore, it is possible that monetary reward is simply not an effective incentive when it comes to regulation of cognitive resources, even if those resources are handled in a rational manner in response to other experimental factors. Future studies could try to manipulate motivation through other kinds of incentives, such as coupling performance with curiosity fulfillment [48, 49] or time penalties.

As is always the case with null results, one possibility is that the absence of effects was due to not having found the "right" experimental conditions to detect the hypothesized effects. We consider multiple experimental design factors that seem to be particularly relevant in this respect. First, one could speculate that the bonuses may have been too small to cause an effect. We believe this to be unlikely, as subjects could increase their total payout by up to a factor 11 if they performed well compared to performing poorly (from $1 to $11 in Experiment 1). Second, subjects might not have had the bonus payments strongly enough on their minds when performing the task. While this explanation could be plausible in Experiment 1—where subjects were informed about the bonus only at the very beginning of the experiment—it seems implausible in Experiments 2 and 3, where they were regularly reminded of it. Third, we may inadvertently have biased our subject sample to "over-performers", by only recruiting subjects who had a high approval rate on the Amazon Turk. The desire to maintain a high approval rate may have worked as a strong incentive for these subjects to perform well, regardless of the amount of performance-related bonus they could earn. However, we know of at least two other studies that have found effects of monetary reward on performance in Amazon Turk subjects with high approval rates, albeit using non-memory tasks [50, 51]. Another possibility is that the absence of any effects was due to using a stimulus type that VWM mechanisms may not be well adapted to. There is increasing evidence that VWM capacity for meaningful, real-life objects is different from VWM capacity for the simple kind of stimuli that typically found in laboratory experiments, like the oriented bars used here [52, 53]. It could be that cognitive control over VWM capacity is limited to stimuli that the system is more adapted too, such as images of natural objects.

The literature on whether total VWM capacity can be manipulated experimentally is small and has thus far produced results that seem to be mixed. Several of those studies have used paradigms similar to the one used here. A recent study by Brissenden et al. [54] tested for effects of reward in an orientation recall task with a set size of two. One difference with our task is that one of the items in their experiment always had a higher probing probability than the other one. Subjects were informed about the probabilities by either a pre-cue (presented before the stimulus) or a retro-cue (presented after the stimulus). The authors found that reward had a beneficial effect on how subjects prioritized cued items over uncued ones, in the sense that rewards decreased recall error for cued items and increased recall error for non-cued item. However, consistent with the results presented here, they found no evidence for an effect of reward on total memory capacity. In another recent by ourselves [55], we 'gamified' the continuous-recall paradigm as a way to manipulate subject motivation. While this manipulation affected both self-reported motivation scores and pupil sizes, also in that study we found that VWM performance was unaffected by any of the experimental manipulations. In a final study that used the orientation recall paradigm, Manga et al. [56] tested for effects of monetary reward on memory precision in younger and older adults. One possibly important difference with the previous three studies is that they presented the stimuli sequentially and with a relatively long duration (800 ms per stimulus, compared to 50 to 200 ms for the entire stimulus display in the other three studies). They found evidence for a positive effect of reward on

memory precision in the younger adults. However, the effect size was small and it did not replicate in the group with older subjects.

Possibly the most convincing evidence that VWM performance can be manipulated experimentally comes from an earlier study by [26]. However, their results are difficult to compare directly with the other studies discussed here, because their experimental paradigm different in several important ways. First, they used color stimuli instead of orientations. Second, and more important, their experimental manipulation consisted of varying the instructions ("focus on a subset", "focus on all items", "simply do your best") rather than monetary reward or motivation. Third, and equally important, they used a full-display change-detection task, which has a more complicated decision stage [57] than a single-probe continuous-recall task. Hence, there are two ways in which their subjects may have improved their performance: by increasing VWM capacity or by using a better decision strategy. One possible explanation of their results—which would reconcile them with the seemingly contradicting results discussed above —is that subjects who were instructed to "focus on all items" may have improved their performance by using a better comparison strategy (between the first and second display) at the decision stage, rather than by increasing their VWM capacities.

In conclusion, the present study adds to a small but growing literature on the question of whether VWM performance can be manipulated experimentally. Taken together, the current literature suggests that performance on VWM tasks *can* be improved through experimental manipulations, but only due to effects on decision strategies—VWM capacity itself seems to be immutable. This apparent immutability stands in contrast with the flexibility found elsewhere in the system, as well as with rational principles of resource utilization, but it is currently the theory that finds strongest support in the empirical data.

## Supporting information

**S1 Appendix.**
(DOCX)

## Author Contributions

**Conceptualization:** Ronald van den Berg, Qijia Zou, Wei Ji Ma.

**Data curation:** Qijia Zou, Yuhang Li.

**Formal analysis:** Ronald van den Berg, Qijia Zou, Yuhang Li, Wei Ji Ma.

**Funding acquisition:** Wei Ji Ma.

**Investigation:** Ronald van den Berg, Qijia Zou, Yuhang Li.

**Methodology:** Ronald van den Berg, Qijia Zou, Yuhang Li, Wei Ji Ma.

**Project administration:** Wei Ji Ma.

**Resources:** Wei Ji Ma.

**Supervision:** Ronald van den Berg, Wei Ji Ma.

**Validation:** Ronald van den Berg.

**Visualization:** Ronald van den Berg.

**Writing – original draft:** Ronald van den Berg, Qijia Zou, Wei Ji Ma.

**Writing – review & editing:** Ronald van den Berg, Qijia Zou, Yuhang Li, Wei Ji Ma.

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
