## [Decision Letter · Decision Letter 0]

24 Oct 2022

PONE-D-22-02333No effect of monetary reward in a visual working memory taskPLOS ONE

Dear Dr. van den Berg,

Thank you for submitting your manuscript to PLOS ONE. After careful consideration, we feel that it has merit but does not fully meet PLOS ONE’s publication criteria as it currently stands. Therefore, we invite you to submit a revised version of the manuscript that addresses the points raised during the review process.

We look forward to receiving your revised manuscript.

Kind regards,

David K Sewell

Academic Editor

PLOS ONE

Journal Requirements:

“This research was supported by grant 2018-01947 from the Swedish Research Council to R.v.d.B, training grant R90DA043849-03 from

National Institutes of Health to Q.Z., and grant R01EY020958-09 from

National Institutes of Health to W.J.M.

“This research was supported by grant 2018-01947 from the Swedish Research Council to 405 R.v.d.B, training grant R90DA043849-03 to Q.Z., and grant R01EY020958-09 to W.J.M”

“This research was supported by grant 2018-01947 from the Swedish Research Council to R.v.d.B, training grant R90DA043849-03 from

National Institutes of Health to Q.Z., and grant R01EY020958-09 from

National Institutes of Health to W.J.M.

Additional Editor Comments (if provided):

Dear Dr van den Berg,

Thank you submitting your manuscript to PLOS ONE. Let me first apologize for the frankly unacceptable delay on engaging the review process with your submission. Upon reviewing the submission history, I was shocked to see how long the journal took to resolve the issue in assigning an editor to the manuscript. As it turns out, the editor on your original submission had left the journal, which I suspect contributed partly to the delay, but by no means excuses it.

Let me provide some details, from what I can piece together. After your initial submission in January, it appears that the journal sought out the original editor, who had left the journal. This was followed by an extended period of inaction with replacement editors only being sought out from late July onwards. I was approached to serve as a new editor on your submission in early August.

I initially approached the two original reviewers, but neither were available to assess your revision. To try and provide some sense of editorial continuity on the submission, I advised the new reviewers about the existing reviews along with your detailed responses to them.

Having now heard back from these two (new) expert reviewers, I am pleased to relay to you that their impressions of the manuscript are highly positive. I have also read through the manuscript and share their general outlook on the work. You will see their comments appended below.

With regards to the previous round of reviews, I think you have clearly addressed all of the major concerns raised by the initial reviewers. The potentially outstanding point being Original Reviewer 2’s question about examining the effect of intrinsic motivation on VWM performance as a function of extrinsic motivation—however, I too was unable to decipher their question, and am happy to leave it at that.

Turning to the comments of the two new reviewers, you will see that their (relatively minor) criticisms overlap with regards to discussing some of the larger theoretical implications of your results. I share their view on this point. The general issue is that much of the Discussion is spent addressing potential study limitations and ruling out alternative accounts. Although there is value in the points you raise, there is relatively little in the way of linking your findings back to existing issues in the literature. For example, Reviewer 1 raises the question of whether your results constitute a falsification of the resource rational framework, or simply limits its scope. Reviewer 2 raises similar concerns along these lines in regards to how your results appear difficult to reconcile with the findings of Bengson and Luck (2016). Clearly, results in the literature are mixed on the matter of flexible allocation of attention at the task level. I do not think the issues must be resolved here—nor do I think they can be, given the current state of play—but some comment on summarizing motivational factors that are and are not implicated in improving VWM performance would be ideal. Reviewer 2 provides some suggestions that are relevant to this that should be addressed.

I also sympathize with Reviewer 1’s view that the theoretical led was buried in relation to resource rational models. I found that the early slots vs. resources was appropriate in contextualizing the genesis of the current work and related ideas, and am happy for this to be left as is. However, I do strongly agree with the reviewer that more could be done to communicate how the resource rational model necessarily predicts a dependency between motivation (and intuitively monetary incentive; though note Reviewer 2’s comments on Camerer and Hogarth’s, 1999, earlier review) and performance. Bringing the model prediction figure in from the appendix to the introduction would be useful in this regard.

Overall, I think it is clear that your manuscript meets the main requirements for publication in PLOS ONE. Before accepting the manuscript for publication though, I would like to see the points raised by the reviewers addressed in a (minor) revision of the manuscript. I am confident that all of the questions/concerns raised by the reviewers can be addressed textually, and so I anticipate being able to make a final decision on a revision without sending the manuscript out for review again.

Thank you again for submitting your manuscript to PLOS ONE. I apologize for the delays on getting reviewer feedback to you, but I hope that the path forward is clear and administratively efficient.

Your sincerely,

David K Sewell

Reviewers' comments:

Reviewer's Responses to Questions

**Comments to the Author**

1. Is the manuscript technically sound, and do the data support the conclusions?

Reviewer #1: Yes

Reviewer #2: Yes

2. Has the statistical analysis been performed appropriately and rigorously? 

Reviewer #1: Yes

Reviewer #2: Yes

3. Have the authors made all data underlying the findings in their manuscript fully available?

Reviewer #1: Yes

Reviewer #2: Yes

4. Is the manuscript presented in an intelligible fashion and written in standard English?

Reviewer #1: Yes

Reviewer #2: Yes

5. Review Comments to the Author

Reviewer #1: I think this work convincingly highlights important boundary conditions for resource allocation in visual short-term memory, and I recommend it for publication. I have a few very minor suggestions for changes.

1) Currently, the theoretical contribution of this paper seems a little bit buried. I suggest a few changes below. I think these revisions would be nice to have, but are not essential, and I leave implementing them to the discretion of the authors.

First, I don’t quiet see why the authors lead with the ‘slots’ versus ‘resources’ debate, which is dropped after the first paragraph and – as the authors note in their response to a reviewer – ‘is largely tangential’ and a ‘can of worms’.

I think one alternative is to take out this paragraph and place more focus on the ‘resource-rational’ framework of visual memory. As part of this revision, I think bringing the model predictions from Appendix 1 to the introduction would be helpful.

Also, if it is true that explicit (monetary) incentives don’t affect visual short-term memory resource allocation – is this a falsification of the resource-rational framework or can it be accommodated to capture these differences? Perhaps I missed it, but this does not seem to be explicitly discussed even though the framework is brought up in the introduction.

At a minimum, it seems like some discussion of these theoretical takeaways would be nice. Currently, the Discussion/Conclusion sections read more like a disjointed list that rules out alternative explanations. I think supplementing it with some higher-level interpretation of the results-- one that is contextualized within existing frameworks/models of visual short-term memory – would elevate the theoretical contribution of the article and make it feel more integrated and self-contained.

2) There are a few minor typos--there may be more, but I caught the following: 3rd sentence of discussion E3 “is was based on 9 randomly selected trials” should be “it was based on 9 randomly selected trials”; last sentence of article “but monetary reward does not may not be one of them” should be “but monetary reward may not be one of them”.

Reviewer #2: The submitted manuscript examines three experiments testing the

relationship between visual memory performance in an online continuous

report task. Overall, I think the paper is clear and admirably

concise. I think the introduction of a third experiment with a

within-subjects manipulation helps demonstrate the generality of the

conclusion here.

I have one concern, which may necessitate a textual change, and two

questions.

1. The discussion was too focused on limitations of the study.

This is partly a stylistic difference (I just feel that such a

negative discussion is both a drag on the paper and a missed

opportunity to draw theoretical links) and partly a sense that the

paper did not fully engage with its theoretical context.

There are interesting questions raised by the current study when

examined in the context of, say, the Bengson and Luck (2016)

result. I don't, for instance, know how to reconcile the idea that

participants can be instructed to pay attention to all of the

stimuli and appear to perform better under those instructions, and

yet not see any effect of monetary reward on performance. The

Bengson and Luck result undermines a large part of the enterprise

to take accuracy directly and estimate an invariant capacity

coefficient, so I think this work is highly useful beyond the

confines of the resource rational model.

The next two points are ones that I think may also warrant

consideration in the discussion, but it is up to the authors how

deeply they wish to engage with them (or whether to engage with

them at all).

2. Optimising performance over many trials may be distinct from

optimising performance in a single trial.

While I appreciate the decision to use two forms of reward

computation in the final experiment, both randomly selecting a

subset to apportion a reward and summing very small rewards over a

large number of trials require the participant to be vigilant for

the entire experiment. The strict resource rational model may

predict that the total amount of mnemonic/attentional resource can

be flexibly allocated on any time scale, but a slightly more

relaxed version of the model could suggest that this level of

control may only be possible on smaller time scales. The literature

cited in the introduction, where precision increases for elements

in a memory array that are rewarded more heavily than others, only

implies that the distribution of performance is asymmetric not that

the total amount changes (this is clear in the introduction), and

that this occurs on a shorter time scale.

In attempting to reconcile these results with the current study

there are three possibilities I can think of: 1. Participants can

control both the resource amount and distribution, but only on

shorter time scales; 2. Participants can control the distribution

of resources but not the total amount; 3. Participants can control

the amount of resources but monetary rewards may not have much of

an effect.

The first possibility has a history in the literature on vigilance

and arousal. The last option is possible given that increasing

performance and motivation in participants through the use of

monetary rewards is fickle at best. Camerer and Hogarth's (1999; J.

Risk and Uncertainty) review of monetary rewards in (mostly

economics) experiments found many have no effect on performance,

and some have a negative effect.

3. What are the implications for models of VSTM performance that

distinguish memory capacity from attentional control?

The primary implications drawn from the experiments focus on

resource rational models of VSTM performance. Another class of VSTM

theory that this bears upon directly are those that distinguish

between transient attentional control and an upper bound on memory

capacity (see, e.g., Adam, Mance, Fukuda, and Vogel, 2015; J. Cog.

Neuro.). It seems difficult to reconcile that performance is

bounded both attentional control and an invariant VSTM capacity

limit, and that both are insensitive to any effect of monetary

rewards.

Minor points

1. The variable precision model was estimated, but neither the

predictions nor the MLE/MAP parameters were presented. Even though

it's not the direct aim of the current study, it is useful to

present all of the details surrounding the fits even if just in an

appendix.

2. The cited Morey et al. (2011) result has antecedents in the work of

Bonnel and Hafter (1998; Percept Psychophys.) and, prior to that,

by Taylor, Lindsay, and Forbes (1967; Acta Psychologica).

6. PLOS authors have the option to publish the peer review history of their article (what does this mean?). If published, this will include your full peer review and any attached files.

Reviewer #1: No

Reviewer #2: No

---

## [Author Response · Author response to Decision Letter 0]

21 Dec 2022

Please see 'Responses to Reviewers'

---

## [Editor Report · Decision Letter 1]

26 Dec 2022

No effect of monetary reward in a visual working memory task

PONE-D-22-02333R1

Dear Dr. van den Berg,

We’re pleased to inform you that your manuscript has been judged scientifically suitable for publication and will be formally accepted for publication once it meets all outstanding technical requirements.

Kind regards,

David K Sewell

Academic Editor

PLOS ONE

Additional Editor Comments (optional):

Dear Dr van den Berg,

Thank you for submitting your revised manuscript. I have reviewed the changes you have made in response to the comments of the reviewers and am satisfied that you have addressed all of their points. Regarding Reviewer 2's "optional" comment that was left unaddressed, my sense is that the additional discussion you provided on the distinction between decision strategy and VWM capacity per se goes some way to speak to their comment. The idea of graded changes in attentional control---potentially related to "task-level" resource allocation by serving a gatekeeper role in governing access to total resources---seems to align well with the question of whether people can adjust capacity in response to certain incentives. However, given your specific focus on resource rational models, I do not think it is necessary to further expand the General Discussion to bring in this additional literature.

In sum, I am very happy to accept your revised manuscript for publication in PLOS ONE. Congratulations! I thank you for your efforts in attentively addressing the comments of all the reviewers throughout the process.

Yours sincerely,

David K Sewell
---

## [Editor Report · Acceptance letter]

3 Jan 2023

PONE-D-22-02333R1 

No effect of monetary reward in a visual working memory task 

Dear Dr. van den Berg:

I'm pleased to inform you that your manuscript has been deemed suitable for publication in PLOS ONE. Congratulations! Your manuscript is now with our production department. 

Kind regards, 

on behalf of

Dr. David Keisuke Sewell 

Academic Editor

PLOS ONE